# Assessing Lung Fibrosis with ML-Assisted Minimally Invasive OCT Imaging

**DOI:** 10.3390/diagnostics14121243

**Published:** 2024-06-13

**Authors:** Rebecca Steinberg, Jack Meehan, Doran Tavrow, Gopi Maguluri, John Grimble, Michael Primrose, Nicusor Iftimia

**Affiliations:** 1Biomedical Engineering Department, Tufts University, Medford, MA 02155, USA; rebecca.steinberg@tufts.edu (R.S.); jack.meehan@tufts.edu (J.M.); doran.tavrow@tufts.edu (D.T.); 2Physical Sciences Inc., Andover, MA 01810, USA; gmaguluri@psicorp.com (G.M.); jgrimble@psicorp.com (J.G.); mprimrose@psicorp.com (M.P.)

**Keywords:** idiopathic lung fibrosis, polarization sensitive optical coherence tomography imaging, machine learning

## Abstract

This paper presents a combined optical coherence tomography (OCT) imaging/machine learning (ML) technique for real-time analysis of lung tissue morphology to determine the presence and level of invasiveness of idiopathic lung fibrosis (ILF). This is an important clinical problem as misdiagnosis is common, resulting in patient exposure to costly and invasive procedures and substantial use of healthcare resources. Therefore, biopsy is needed to confirm or rule out radiological findings. Videoscopic-assisted thoracoscopic wedge biopsy (VATS) under general anesthesia is typically necessary to obtain enough tissue to make an accurate diagnosis. This kind of biopsy involves the placement of several tubes through the chest wall, one of which is used to cut off a piece of lung to send for evaluation. The removed tissue is examined histopathologically by microscopy to confirm the presence and the pattern of fibrosis. However, VATS pulmonary biopsy can have multiple side effects, including inflammation, tissue morbidity, and severe bleeding, which further degrade the quality of life for the patient. Furthermore, the results are not immediately available, requiring tissue processing and analysis. Here, we report an initial attempt of using ML-assisted polarization sensitive OCT (PS-OCT) imaging for lung fibrosis assessment. This approach has been preliminarily tested on a rat model of lung fibrosis. Our preliminary results show that ML-assisted PS-OCT imaging can detect the presence of ILF with an average of 77% accuracy and 89% specificity.

## 1. Introduction

Interstitial lung disease (ILD) is a collective term for life-threatening illnesses that cause inflammation and fibrosis (scarring) of lung tissue [1]. ILD is a chronic progressive condition that causes worsening fibrosis over time. Pulmonary fibrosis (PF), also called idiopathic pulmonary fibrosis (IPF), is an advanced stage of ILD when the air sac in the lungs (alveoli) becomes scarred and stiff (see Figure 1), making it difficult to breathe and get enough oxygen into the bloodstream [2]. IPF is a devastating condition that carries a prognosis worse than that of many cancers. As such, it represents one of the most challenging diseases for chest physicians [3,4,5,6,7,8].

ILD causes permanent lung damage. It is not usually possible to fix lung scarring that has already occurred. However, if diagnosed in an early stage, various treatments can be used to slow down the progression of the disease and improve symptoms. For example, corticosteroids such as prednisone can help to reduce inflammation. There are FDA-approved anti-fibrotic agents such as nintedanib (Ofev) and pirfenidone (Esbriet), which slow down lung scarring [9]. Furthermore, cytotoxic drugs like azathioprine, cyclophosphamide, and mycophenolate mofetil (Myfortic, Cellcept) slow down lung fibrosis by suppressing the immune system [10]. The type of selected therapy is a function of the ILD severity. Therefore, it is very important to precisely assess ILD severity.

The diagnostic process for ILD is complex and heavily relies on the pulmonologist’s capability to integrate clinical, laboratory, radiologic, and/or pathologic data. Therefore, ILD misdiagnosis is still common, resulting in patient exposure to costly and invasive procedures, as well as inefficient use of healthcare resources [11]. Genetic testing is now available [12], but still there is not sufficient clinical evidence that this type of diagnosis is sensitive and specific. Furthermore, it cannot be used to differentiate between stages of ILD, and therefore biopsy is still needed for a final evaluation. Video-assisted thoracoscopic surgery (VATS) under general anesthesia is usually performed to obtain enough tissue to analyze its undistorted morphology and make an accurate diagnosis [13]. This procedure is highly invasive, as it involves placement of several tubes through the chest wall, one of which is used to cut off a piece of lung to send for evaluation. The removed tissue is examined by a histopathologist to confirm the pattern of fibrosis. However, the procedure can potentially lead to ILD exacerbation, and in many cases brings additional risks, including severe pulmonary hemorrhage, acute respiratory failure, and death [14]. Other complications include tissue morbidity, inflammation, and pneumothorax (air leakage into the space between lung and chest wall, which pushes on the outside of the lung and makes it collapse). Pneumothorax is reported to occur in 17–27% of patients when wedge biopsy is performed [15]. Furthermore, VATS patients require hospitalization, which can be extended for many days or weeks if complications occur.

More recently, minimally invasive optical imaging approaches, such as optical coherence tomography (OCT), have assessed lung tissue morphology and changes induced by fibrosis [16,17]. OCT uses a fiber optic catheter that can be deployed through the instrument channel of a bronchoscope. The polarization sensitive version of OCT (PS-OCT) enables the analysis of birefringence properties of fibrous structures of tissue, and thus the determination of the presence of fibrosis [18,19]. Birefringence is formally defined as the double refraction of light in a transparent, molecularly ordered material, which is manifested by the existence of orientation-dependent differences in the refractive index.

PS-OCT has been extensively used for assessing lung fibrosis. It is used to assess excessive collagen accumulation and thus the identification and quantification of fibrosis. A recent study performed by M. Vaselli et al. [20] has shown that ILD features were reliably identified with PS-OCT imaging. Microscopic ILD features were identified on both in vivo and ex vivo PS-OCT images. T. Soldati et al. [21] has also shown that minimally invasive PS-OCT is a safe imaging technique to detect and quantify pulmonary fibrosis. In 49 out of 55 imaged cases, parenchymal birefringence was quantified, ranging from a mean fibrosis score of 2.54% (no to minimal fibrosis) to 21.01% (extensive fibrosis). Nandy et al. [22] demonstrated that PS-OCT can reliably differentiate between interstitial pneumonia (IP) and a non-IP histopathological pattern in fibrotic lung disease. In a 27-patient study, they demonstrated 100% sensitivity and specificity for diagnosing an IP histopathological pattern. Hariri et al. [23] have also shown that PS-OCT can provide reliable measurement of birefringent fibrosis and total collagen content. However, the typical OCT catheters used in these studies do not provide accessibility to deep areas in the lungs, where the fibrosis starts, unless deployed through the bore of a long and difficult to maneuver needle.

In this paper we report a simpler procedure for lung fibrosis assessment based on a percutaneous approach, which enables access to any area of the lung. In contrast to the reported OCT systems, our system uses a short probe (~7 inches in length) with a diameter of 0.9 mm, which can be placed in any region of interest within the lung under ultrasound (US) or CT guidance. The use of a small-diameter probe significantly minimizes biopsy complications in comparison to the wedge biopsy approach. The minimally invasive OCT probe enables the collection of high-quality PS-OCT images, which can be analyzed with a custom ML-based algorithm to assess IPF presence and level of severity. We performed a preliminary evaluation of this approach in a rat model of lung fibrosis. The overall performance of the ML model was similar to that of the humans performing the same classification tasks. Specifically, tissue segmentation was excellent, closely mimicking ground truth provided by the human annotations, while 77% accuracy and >89% specificity were obtained for tissue type classification.

## 2. Materials and Methods

**PS-OCT instrumentation**: A polarization-sensitive optical coherence tomography (PS-OCT) instrument was developed and used to assess tissue morphology and birefringence (see schematic in Figure 2). A brief description of this system is presented in the following. A polarization-maintaining (PM) fiber approach is used to decompose light into two orthogonal polarization states, which were processed to extract sample birefringence information using the typical Jones formalism [18]. Unlike typical PS-OCT systems that require the use of two expensive spectrometers, our more cost-effective solution requires only a single spectrometer. This solution is enabled by using a custom encoder-based imaging approach [24]. A fiber optic switch (NanoSpeedTM 1 × 2 Series Agiltron) is used to sequentially send the two orthogonal polarization states of the light to the same spectrometer. With a switching speed of 300 ns, no imaging artifacts were observed. By using an axial scanning speed of ~2 cm/s (2 cm axial scan), the probe only moves 0.4 µm during a 20 µs acquisition time for the two polarization states. This means that the two polarization states are practically collected from the same tissue location. As the lateral resolution of the instrument is ~10 µm, the 0.4 µm overlap error does not create any noticeable loss in resolution.

The instrument uses a 1310 nm light source with a bandwidth of approximately 95 nm, enabling an axial resolution of ~8 µm. This axial resolution supports the detection of smaller tissue features at the cellular level. The light from the broadband source is sent to a polarizer and then split into the sample and reference arms of the interferometer by a 50/50 fiber splitter. Fiber coils are used to separate the parallel and perpendicular states that can beat and generate imaging artifacts. A quarter wave plate oriented at 22.5° is used in the reference arm to obtain circular polarization of the light returning from the reference arm and thus to maximize the fringe signals for both polarization states of the light returning from the reference arm. A 45° Faraday rotator is used in the sample arm to send a circularly polarized light beam to the sample. The light from the 4th port of the splitter (25% of the collected light from each arm of the interferometer) is split into two orthogonal polarization states by a 50/50 polarization beam-splitter and sent to the spectrometer through the fiber optic switch. A polarization controller (PC) is used to balance the power between the two orthogonal polarization states. To avoid polarization mode dispersion (PMD) artifacts [25], no fiber circulators are used in our setup. The acquisition of every OCT reflectivity profile (A-line) is triggered by a linear optical encoder (Model MTE-4, MicroE Systems, Bedford, MA, USA). The incremental movement of the catheter probe is detected by this encoder, which creates a pulse that triggers the spectrometer camera. The same signal is detected by a DAQ counter input to start the OCT data processing loop. The processed OCT signal is inserted into an array that was appended at each incoming trigger signal to form an OCT image. The fringe signals are digitized by a camera link frame grabber and processed in real-time by a Graphical Processing Unit (GPU).

**OCT probe:** A specially designed OCT probe, suitable for tissue investigation through the bore of the biopsy needle, was used in this study. The design of this probe was recently reported [26]. In short, the probe consists of four major parts: probe main body, plunger, encoder, and needle containing the OCT fiber optic catheter (see Figure 3).

The fiber optic OCT catheter plunger is attached to the spring-loaded plunger through a fiber connector (see cross-sectional and transparent views). When pressed, the plunger moves the OCT catheter forward within a custom-made needle. The OCT fiber catheter consists of a single mode (SM) fiber, terminated with a micro lens, polished at 45 deg, to send the light orthogonal to the catheter scanning direction. Thus, the OCT light exiting the needle through a slot made at the tip of the needle scans the tissue. The needle is covered on the slot area by a fluorinated ethylene propylene (FEP) tube to seal the OCT catheter inside the needle and prevent the tissue from catching on the needle.

An optical encoder is attached to the probe holder and used to monitor the movement of the OCT fiber optic catheter relative to an optical scale, which is also attached to the plunger. An electronic circuit, inserted into the probe body, is used to allow for an A-line acquisition only when the plunger is moved. Thus, it blocks false triggers generated by the small vibrations during probe insertion within the tissue before pushing the plunger. This circuit also forms the trigger signal, so it can be reliably sent through a 2 m length mini-USB cable to the instrumentation unit.

**Animal model:** An in vivo study for technology evaluation was performed at the animal facility of University of Massachusetts Lowell (UML). A bleomycin (BM) rat model of lung fibrosis was used [27]. All experiments were performed in agreement with the UML IAUCUC-approved animal protocol—00001349-RN00-AR002. Optical imaging was performed at 7, 14 and 28 days post-BM exposure under anesthesia. Immediately after imaging, the animals were sacrificed, and postmortem histological analysis of lung tissue was performed to study the pathological attributes of pulmonary fibrosis.

The experimental plan consisted of the following steps:

(1) Induction of pulmonary fibrosis: Sprague Dawley rats (7 weeks old, male) were anesthetized using isoflurane and administered a single dose of bleomycin (BM) in saline (2.5 mg/kg) by intratracheal instillation (n = 4 per group). Matched saline control groups were also included with n = 1 to 2/group.

(2) Thoracic surgery: Since the lung of the rat was rather small, surgery was performed under ketamine/xylazine anesthesia to slightly expose the lung and avoid puncturing vital organs during optical biopsy.

(3) In vivo optical imaging: Imaging was performed at time points 7, 14 and 28 days post-BM exposure (see timeline in Figure 4). The probe needle was gently inserted into the main lobe, at ~25 deg angle over a depth of <10 mm, and OCT scans were collected between the respiration cycles. Two (2) measurement sites (one in the left and one in the right lung) were selected; the total duration of imaging was <5 min per animal.

(4) Tissue marking: After removing the OCT probe, a fine needle with India ink was inserted on the OCT path to mark the OCT imaging location for histology correlation purposes.

(5) Euthanasia: Following the optical biopsy, the animals were euthanized using Beuthanasia-D (1 mL/10 lb), the whole lung was purged with saline, and the trachea was sutured to avoid its collapse. Then, the lung was excised, fixed with 10% formalin, and processed for H&E and Masson’s Trichrome staining to identify collagen fibers.

**Data Processing:** A custom image segmentation/ML analysis software (Version 1) was developed to interpret imaging findings and estimate the level of fibrosis invasion: mild, moderate, and severe. These criteria were developed in agreement with the well-established Ashcroft scale [28], as summarized in Table 1.

Examples of OCT-histology correlated images are shown in Figure 5, Figure 6 and Figure 7. Both the reflectance and the phase retardance birefringence images are presented. Gradual thickening of the alveoli walls may be noticed with the invasion of fibrosis, as well as reduction in the size of the alveoli. Furthermore, the advancement of fibrosis within the animal lung may be noticed through the increased tissue birefringence (see color scale) as the walls of the alveoli become stiffer.

To make the differentiation possible between normal and fibrotic lungs, the software was conceived to capture the average alveoli size, different percentiles of the alveolar size distribution (10th, 25th, 33rd, 50th, 67th, 75th, and 90th), the average wall thickness (AwT), different percentiles of the wall thicknesses, the range of the alveoli size (As)/alveoli wall thickness, the standard deviation of the alveoli size/wall thickness, and the area of tissue within an OCT scan with increased birefringence or fibrosis invasion status (FiS).

**Data Segmentation:** First, to establish a ground truth for our experimental model, the images were first manually annotated in agreement with histopathology and separated into three classes: normal (healthy), mild, and severe fibrosis. Each identifiable alveoli (alveoli with an area greater than two times the speckle size and alveoli that were not statistical outliers) had their areas and wall thickness measured. These annotations were compared to the computer-generated metrics using ANOVA and Tukey’s honest significant difference. Next, traditional image analysis was conducted to gather data that was fed into a ML classifier. Each image was segmented into smaller sub-images to increase the data set size, as shown in Figure 8.

Although more than one image was taken from each measurement site, a single representative image per site was c = considered, as the images were very much alike. After each image was segmented, the total number of images increased from 16 (7 advanced fibrosis, 6 mild fibrosis, and 3 normal) to 54 (20 advanced fibrosis, 20 mild fibrosis, and 14 normal), as shown in Table 2. Synthetic minority oversampling technique (SMOTE) was then applied to the split images to obtain a fully balanced dataset.

The segmentation/ML code was written in Python. The Python script filtered the birefringence images with different color masks (see Figure 9) and calculated the percentage of pixelation. An HSV model (hue, saturation, and value) was used for image processing. HSV 0–20 was used for red, 20–40 was used for yellow, 50–80 was used for green, and 90–120 was used for blue. 

To determine the alveolar metrics, the program first processed the images by highlighting the walls edges with the white top-hat function of the scikit-image module, then performed edge smoothing and noise reduction with gaussian blurring, followed by thresholding to convert grayscale into binary, coloring the alveolar area as black and surrounding walls and tissue as white. Next, binary erosion was used to remove small perturbations on the boundaries, before dilating the image to return the boundaries to their correct size with a smoother outline and applying canny edge detection to finalize the alveolar outlines. The alveolar areas and walls were measured by first identifying the centers, and then by calculating the space inside the boundary for the area and by calculating the radial distance between an edge of the current alveoli and the start of another alveoli. Finally, a pandas data frame was used to collect the alveolar metrics and color compositions obtained after image processing. Each row represented one image, and columns represented the different percentiles of the alveolar size and wall thickness, percent color pixelation values from the birefringence images, and the FiF, which was calculated as the ratio of the summation of the percentiles of each color filter to the total image area.

## 3. Results

Comparisons between the algorithm detected areas and wall thicknesses versus the ground truth areas and wall thicknesses are shown in Figure 10.

Differences between the ground truth and algorithm results were all statistically significant, except for the alveolar areas of the advanced fibrotic tissue (one way ANOVA *p*-value of 0.1990). The advanced wall thicknesses had a *p*-value of 0.0002. The *p*-values of the mild areas and wall thicknesses were 0.0035 and 3.4742 × 10^−5^, respectively. The *p*-values of the normal areas and wall thickness were 0.0002 and 1.2594 × 10^−9^, respectively.

The average alveolar size for advanced, mild, and normal samples were 81.5 pixels, 88.6 pixels, and 131.8 pixels, respectively, as shown in Figure 11. Additionally, the differences between the means advanced-versus-normal and mild-versus-normal groups were statistically significant as determined by the Mann–Whitney test. The *p*-values were 0.0035 between the normal and advanced samples and 0.0004 between the normal and mild samples. Figure 9 also shows that the average wall thicknesses for advanced, mild, and normal samples were 24.8 pixels, 33.8 pixels, and 22.8 pixels, respectively. The differences between the advanced versus mild groups and the mild versus normal groups were statistically significant as determined by the Mann–Whitney test. The *p*-values were 0.0043 between the advanced and mild samples, and 0.0006 between the normal and mild samples.

The differences in color averages between each disease state were analyzed as well (see Figure 12). It was found that the average percentages of red pixels for the advanced, mild, and normal samples were 4.32%, 2.69%, and 0.83%, respectively. All these values were found to be statistically significant with a Mann–Whitney test (advanced to norm = 1.4980 × 10^−7^, advanced to mild = 0.0060, mild to norm = 3.9285 × 10^−5^). The average percentages of yellow pixels for the advanced, mild, and normal samples were 6.50%, 5.86%, and 2.88%, respectively. All these values were also found to be statistically significant (advanced to norm = 1.0529 × 10^−6^, advanced to mild = 0.02564, mild to norm = 9.5525 × 10^−5^).

The average percentages of blue pixels for the advanced, mild, and normal samples were 2.66%, 4.75%, and 3.70%, respectively. The advanced blue mean was significantly lower than the mild and normal means (advanced to norm = 0.0300, advanced to mild = 0.0003). The average percentages of green pixels for the advanced, mild, and normal samples were 6.67%, 6.88%, and 6.47%, respectively, with no statistical significance between these differences.

**Model Selection**: PyCaret was used to narrow down potential model types. The top five suggested models from PyCaret were manually tested. As shown in Figure 13, the gradient boosting classifier performed the best, accurately predicting 41 out of 60 samples. The logistic regression classifier performed the second best, as it accurately predicted 40 out of 60 images. The basic decision tree performed the third best, with 39 out of 60 images being correctly predicted. Fourth, the bagged decision tree performed the worst, as it only predicted 37 out of 60 samples correctly. Finally, the Ada boost classifier predicted 35 out of 60 samples correctly, making it the least accurate model.

**ML Training and Validation:** Since the gradient boosting classifier had the highest accuracy, it was chosen as our final model (see Figure 14). The model was built and validated using 10-fold validation. From the 10-fold validation the precision, recall, specificity and F1-score were calculated. These values were shown in Table 3.

Finally, the model was interrogated to reveal the important features when determining the IPF classification. These features were the alveoli area metrics (namely the 67th percentile, the standard deviation, and range), the wall metrics (namely the 25th and 75th percentiles, all the birefringence metrics, and the FiF). This is shown in Figure 15.

## 4. Discussion

This study has shown that PS-OCT may be a suitable technique for assessing lung IPF. Furthermore, it has shown that gradient boosting classifier (GBC) provides the most accurate results for disease status assessment when combined reflectance and birefringence images are analyzed. The GBC performed the best out of all the proposed classifiers, with an accuracy of 77% when using a 10-fold cross validation.

Initially, the analysis was performed on a dataset that was heavily unbalanced, with very few normal tissue samples to create meaningful models. Thus, image segmentation and SMOTE were performed to balance the dataset, as shown Table 2. Segmenting images by the number of alveoli allowed for normalized metrics, such as average alveolar, to be computed with an equal amount of predictive power. By further balancing the dataset through SMOTE, it was ensured that all classes could have equal predictive power in the final classifier.

The increased performance of the GBC classifier might be due to its ability to analyze all the metrics used in this study: the color percentages showing increased birefringence, the alveolar area, and wall thickness metrics. Figure 15A showed the key features for disease status classification: red composition, yellow composition, and the 75th percentile of the wall thickness measurements. When further interrogated in the dataset, the percent of red pixels and percent of yellow pixels both had clear separations between each of the fibrotic classes, which is consistent with the expected results. In the birefringence images, yellow and red pixels denoted areas of increased collagen. Additionally, it is known that fibrotic tissue has a high composition of collagen, making it a strong biomarker of IPF [29]. Thus, Figure 9B and Figure 12A show that our dataset adheres to the expected birefringence trends, where an increase in red and yellow pixels correlate with a worse disease state. Taken together, this shows that the GBC classifier was able to make a biologically sound decision regarding the presence of fibrotic tissue using the red and yellow composition metrics. Additionally, the GBC identified trends in alveolar wall thicknesses that are consistent with expectations. With a healthy sample, the alveoli would be large and uniform with short distances between them. In the later stages of the disease, the alveoli tend to be smaller and farther apart [30]. Thus, the GCB model was able to notice this expected difference between the alveolar wall thicknesses. This is clearly seen in Figure 11B when analyzing the wall thicknesses, as the range and 75th percentile of the mild fibrosis images was much higher than that of the normal images.

A notable problem in our study was sample inadequacy due to the limited number of animals used in the pilot study. A small and unbalanced data set makes it difficult to ascertain the true validity of classification by alveolar metrics and birefringence data from our final model. Segmenting our images for every 10 alveoli achieves greater k-fold validation capabilities and more accurate metric distributions but sacrifices the weight of alveolar metrics on classification by increasing the volatility in normal and mild images to that of the advanced ones which average. The contrasting effect that image segmentation has on the two major categories of data may be explained by the fact that fibrotic invasion occurs in the extracellular space whereby collagen measurements are made at thousands of individual pixels, while all alveoli metrics are calculated from 10 data points grouped by proximity, making it likely that each set varies from the population metrics. This reduces the likelihood of correlation between these values and diagnostic accuracy, leaving PyCaret to distinguish the most optimal classification models with a heavy bias on birefringence data, which we see from Figure 13. With a more adequate data set, image splitting would not be necessary. This would hopefully translate into balanced significance of alveolar metrics and birefringence data and potentially a different type of classifier.

While less consequential to the final model, Figure 8 illustrates the disparity in distribution of manually annotated wall lengths and the computational measurements. Even though the distributions were mostly different, the ranges of the algorithm’s results and the ground truth’s results had similar ranges and peaks, especially in the normal and advanced samples. The algorithm was less accurate when measuring the wall thicknesses, as the algorithm had a larger range of reported values than the ground truth annotations. However, the normal sample showed a similar spike in density around a thickness of 10 pixels. These discrepancies could have been caused by the algorithm measuring wall thickness at predetermined angles, which could have resulted in more measurements than a human would have made. If the manual annotations were to be redone, a focus would be made on replicating the exact measurement strategy used by the algorithm as opposed to the common sense ruling that walls are only drawn to nearby alveoli, not in arbitrary directions until another is reached.

It was originally hypothesized that any sort of classification would be highly dependent upon mean alveolar area and wall length, as well as FiF. The model identified the prevalence of yellow and red as being highly impactful parameters, while the mean alveolar area and wall length were less impactful. This lends to two outlooks, being that either the model achieved 77% accuracy despite a lack of accurate alveolar metrics characteristic of the different disease state physiologies, or that this hypothesis is null and these factors are not as important as originally predicted. The ambiguity of machine learning using suboptimal data leaves this question unanswered. However, it is our belief that highly significant metrics exist, and with a more fleshed out data set, our approach would garner more success. The rudimentary segmentation algorithm still does not capture all the alveoli, and improvements can be made on the wall thickness determination. Future work on both of those topics should improve the overall performance. Additionally, it is expected that humans will have larger alveoli than the Sprague Dawley rats, which will also improve the performance of the classifier. Larger alveoli should allow for easier detection by the alveoli detection algorithm. Despite these factors, the model still outperforms current diagnostic alternatives. As such, we maintain the hypothesis that birefringence and alveolar metric data should provide enough information to achieve an IPF diagnosis through computational modeling.

The novelty of this work should not be understated, especially that simpler and easier-to-fabricate OCT probes than those previously reported can be used to provide reliable PS-OCT images and assess fibrosis invasion status. Furthermore, this is the first attempt of using ML to automatically analyze PS-OCT images and assess IPF severity. As far as we know, we are the first to use OCT as the driving classifier, and in contrast, are attempting to create an algorithm that utilizes in situ imaging. Additionally, while other works have also found that an increase in birefringence corresponds with a worse fibrotic disease state [22,23,25], such other studies have not built a machine learning model that could aid in the diagnosis of new IPF cases. Nevertheless, there is a long way ahead until a technology like this could be clinically translated. A finer scale of fibrosis invasion status is needed to fully comply with the Ashcroft scale, while the ML approach used in this study is very preliminary and needs further refinement.

## 5. Conclusions

This paper presents a novel approach for lung fibrosis detection, which may be further perfected and used to change the landscape for idiopathic pulmonary fibrosis diagnosis. Idiopathic pulmonary fibrosis is often left undiagnosed or misdiagnosed as a less serious respiratory condition. Diagnosis currently requires the collection of tissue samples, which takes significant time and exacerbates IPF symptoms—a disease that is already destructive and despairing. There is a need for an IPF diagnostic tool that can achieve the same diagnostic accuracy as invasive biopsy, without harmful effects.

We developed a novel probe for in situ biopsy and showed that it can retrieve morphological and biomechanical changes specific to fibrosis invasion. We extracted a range of features from these images, including the alveolar size and area, alveolar wall thickness, and birefringence makeup of each image. We correlated these features with histological diagnosis. We also developed an ML classification algorithm to analyze the collected data and determine fibrosis invasion severity. A gradient boosting classifier was able to distinguish between advanced fibrosis, mild fibrosis, and normal lungs with over 77% accuracy and over 89% specificity. Considering that this analysis was performed with an unbalanced dataset of images from only 15 animals, we believe the predictive capabilities of our algorithm will only increase as we accrue more data. There are many parameters for the classification algorithm, including random state, learning rate, number of estimators, and max depth, which must be further tuned for the best possible prediction accuracy on a larger data set of images.

## Figures and Tables

**Figure 1 diagnostics-14-01243-f001:**
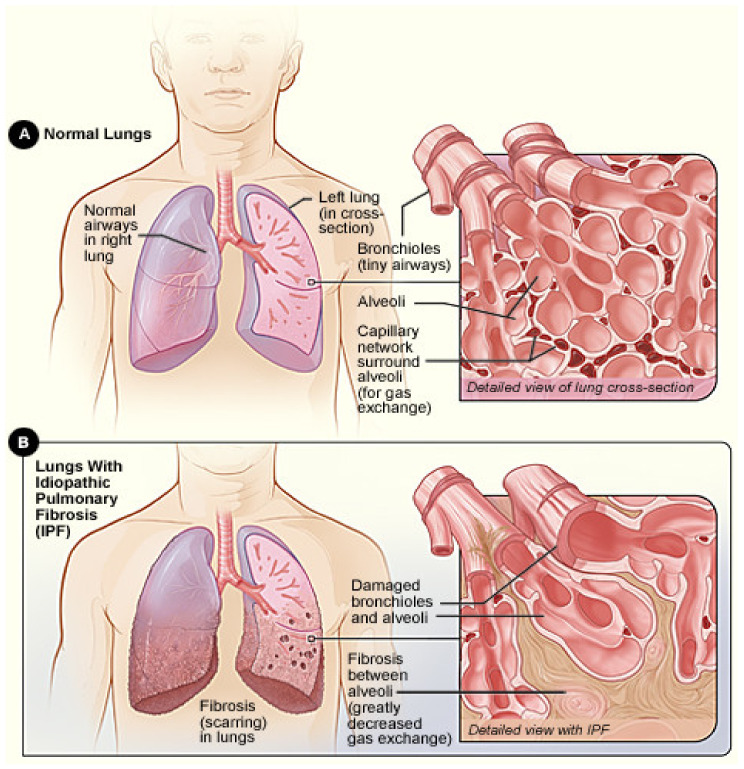
Normal lung vs. lung with IPF. (**A**) shows the location of the lungs and airways in the body. The inset image shows a detailed view of the lung’s airways and air sacs in cross-section. (**B**) shows idiopathic pulmonary fibrosis (scarring) in the lungs. The inset image shows a detailed view of the fibrosis and how it damages the airways and air sacs. [Source: National Heart Lung and Blood Institute (NIH)—https://commons.wikimedia.org/wiki/File:Ipf_NIH.jpg?uselang=en#/media/File:Ipf_NIH.jpg (accessed on 3 June 2024)].

**Figure 2 diagnostics-14-01243-f002:**
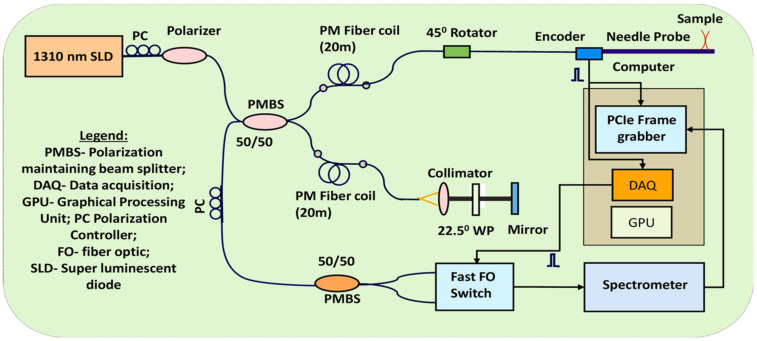
Schematic of the encoder-based PSOCT instrument.

**Figure 3 diagnostics-14-01243-f003:**
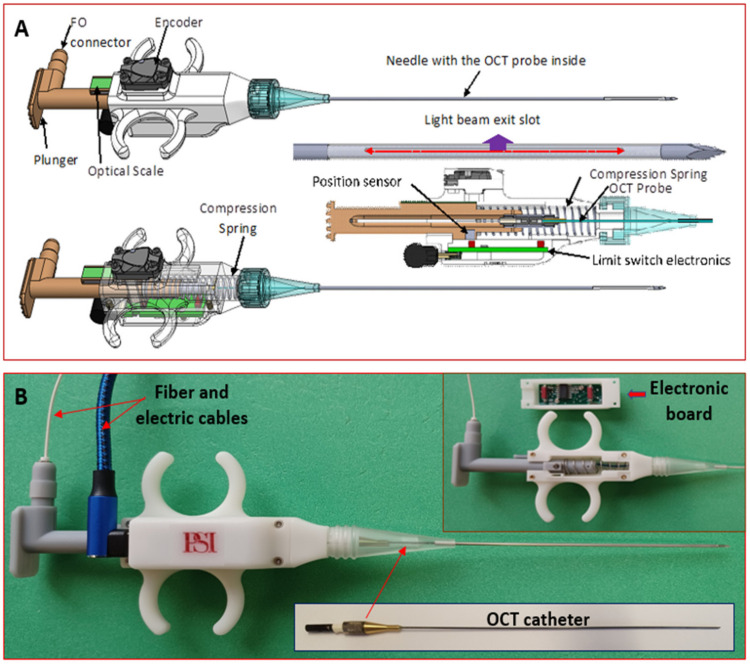
(**A**): CAD design of the biopsy probe. Top—general view; middle—needle and spring-loaded mechanism details; bottom—transparent view. (**B**): Photographs of the optical probe showing inside details.

**Figure 4 diagnostics-14-01243-f004:**
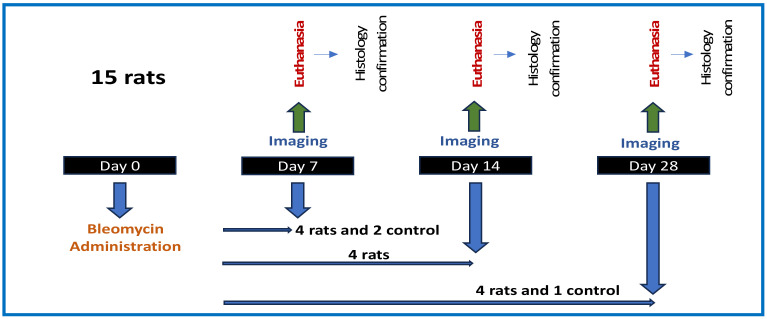
Animal study timeline.

**Figure 5 diagnostics-14-01243-f005:**
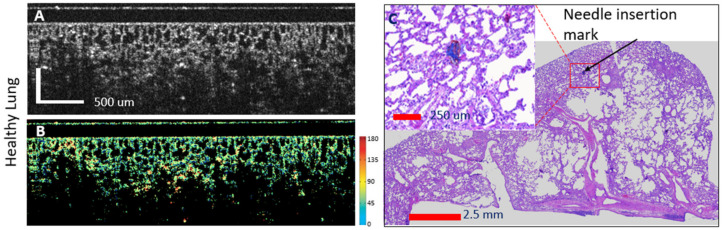
Healthy lung: OCT—histology-confocal correlation and automated assessment of fibrosis presence. (**A**)—reflectance OCT image; (**B**)—birefringence OCT image. Each color indicates a different orientation of the tissue optic axis (right-sided color scale); (**C**)—histology of the lung lobe with magnification of the needle insertion area.

**Figure 6 diagnostics-14-01243-f006:**
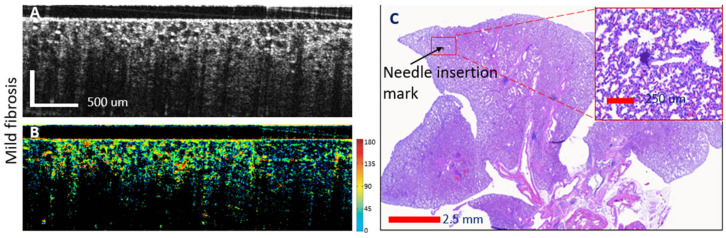
Mild fibrosis lung: OCT—histology-confocal correlation and automated assessment of fibrosis presence. (**A**)—reflectance OCT image; (**B**)—birefringence OCT image. Each color indicates a different orientation of the tissue optic axis (right-sided color scale); (**C**)—histology of the lung lobe with magnification of the needle insertion area.

**Figure 7 diagnostics-14-01243-f007:**
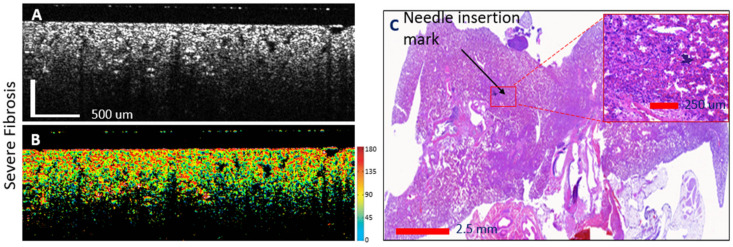
Advanced fibrosis lung: OCT—histology-confocal correlation and automated assessment of fibrosis presence. (**A**)—reflectance OCT image; (**B**)—birefringence OCT image. Each color indicates a different orientation of the tissue optic axis (right-sided color scale); (**C**)—histology of the lung lobe with magnification of the needle insertion area.

**Figure 8 diagnostics-14-01243-f008:**
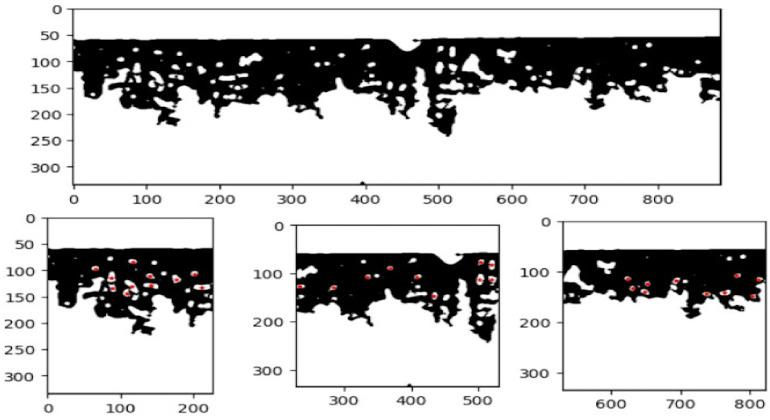
Visualization of image segmentation. The top row shows a processed image. The bottom row shows the same image, separated in patches, which are used in the ML. Alveoli are denoted by red dots. Units are in pixels.

**Figure 9 diagnostics-14-01243-f009:**
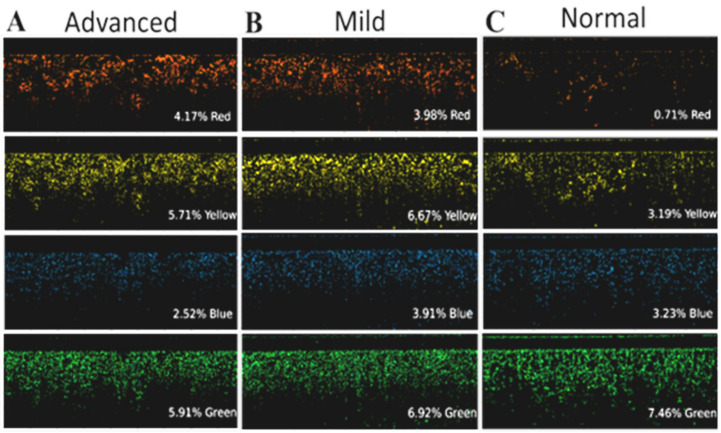
Color segment of the birefringence data based on RGB thresholds. (**A**) Results of a normal sample. (**B**) Results of a mild fibrosis sample. (**C**) Results of an advanced fibrosis sample.

**Figure 10 diagnostics-14-01243-f010:**
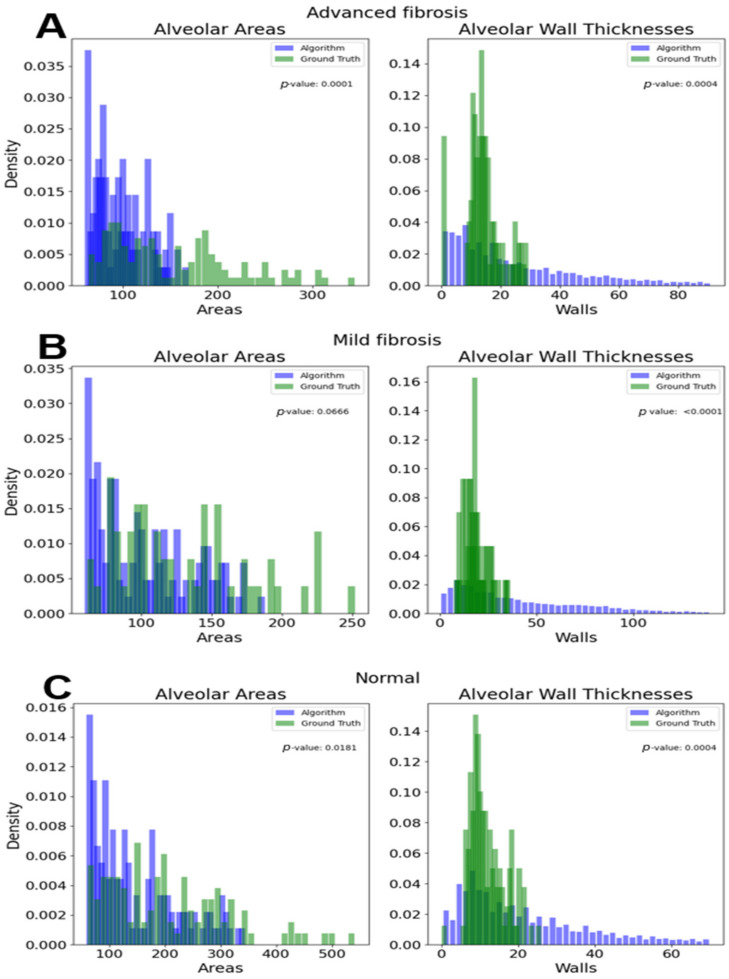
(**A**) The density distributions of all advanced fibrosis samples. (**B**) The distributions of all mild fibrosis samples. (**C**) The distributions of all normal samples.

**Figure 11 diagnostics-14-01243-f011:**
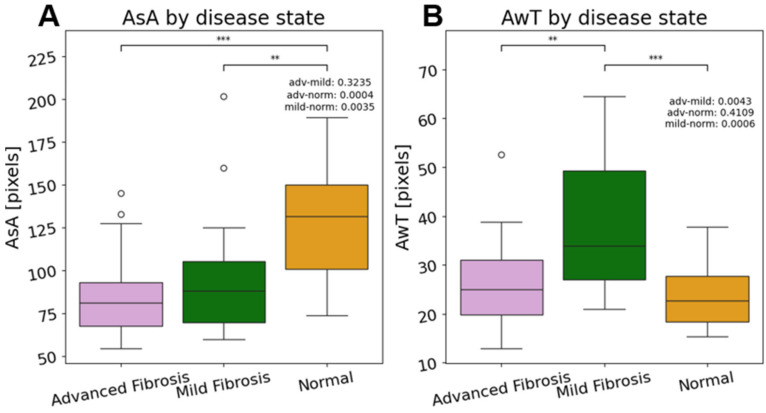
Difference in average alveolar size (AsA, panel (**A**)) and average wall thickness (AwT, panel (**B**)) between the advanced, mild, and normal samples. ** significant difference in STD; *** moderate difference in STD. ** *p* < 0.01; *** *p* < 0.0001.

**Figure 12 diagnostics-14-01243-f012:**
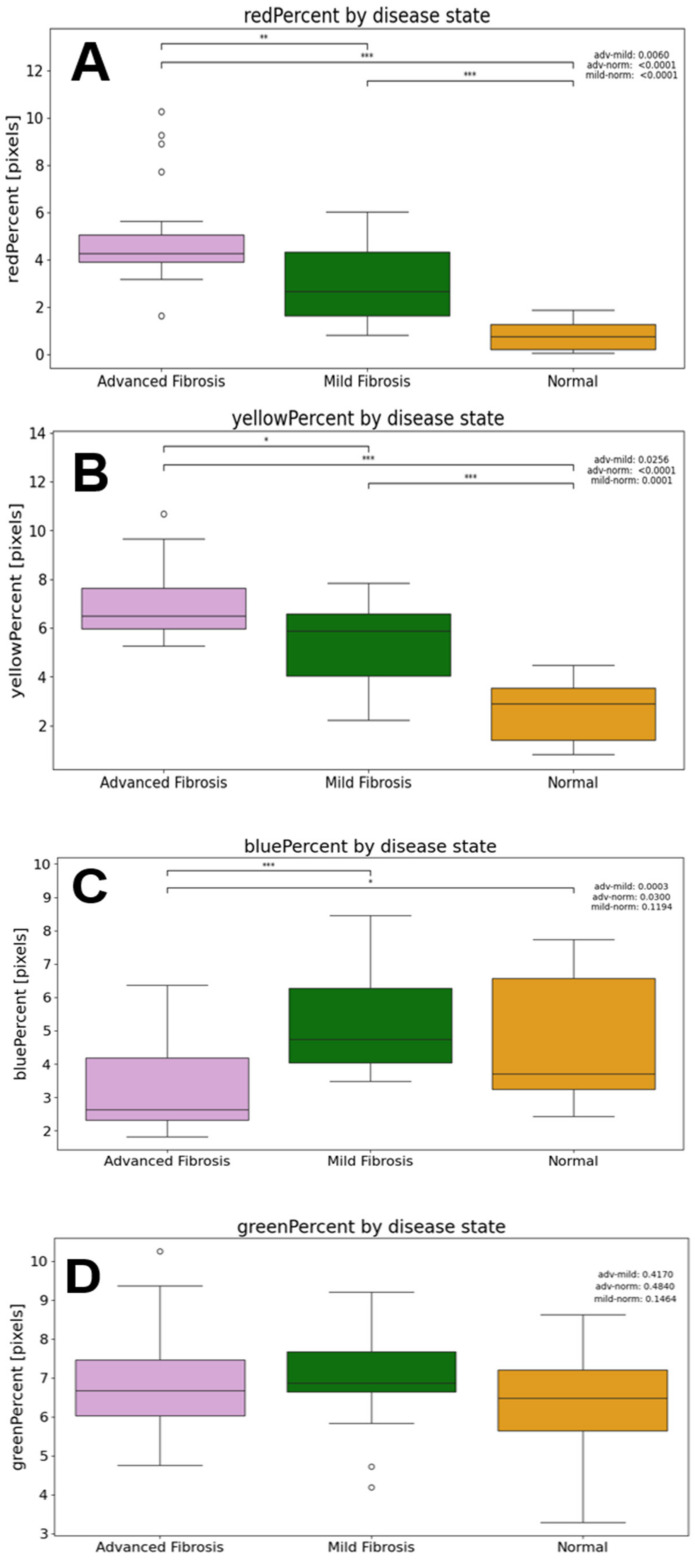
(**A**) Red percent analysis. (**B**) Yellow percent analysis. (**C**) Blue percent analysis. (**D**) Green percent analysis. * *p* < 0.05; ** *p* < 0.01; *** *p* < 0.0001.

**Figure 13 diagnostics-14-01243-f013:**
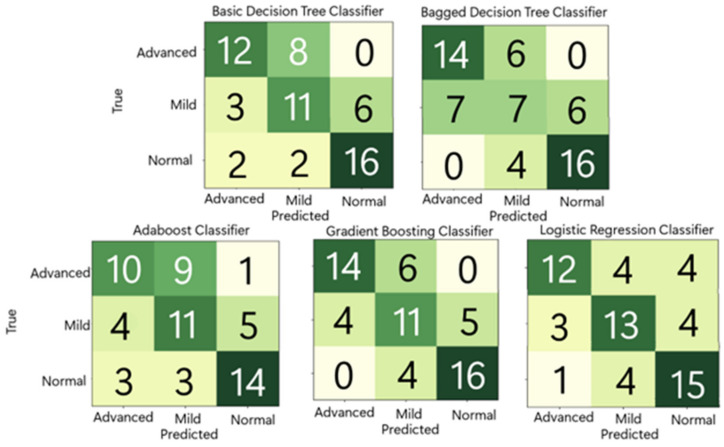
Comparison of the five top performing models, according to PyCaret. Class 0 is advanced fibrosis, class 1 is mild fibrosis, and class 2 is normal.

**Figure 14 diagnostics-14-01243-f014:**
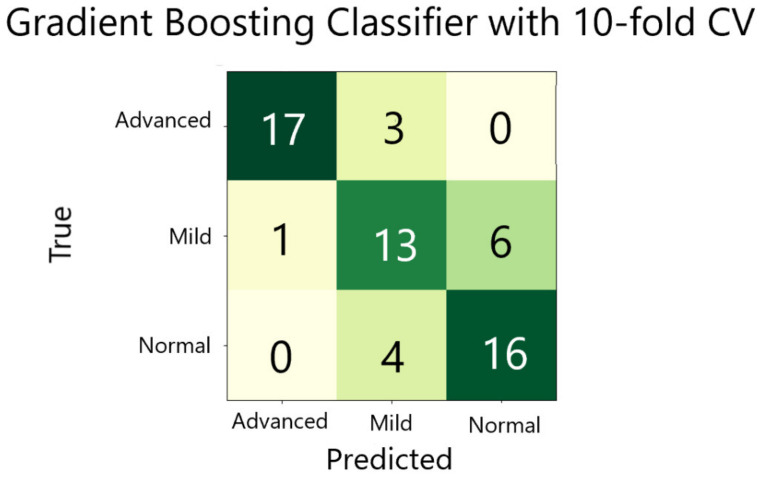
The confusion matrix of the cross validation.

**Figure 15 diagnostics-14-01243-f015:**
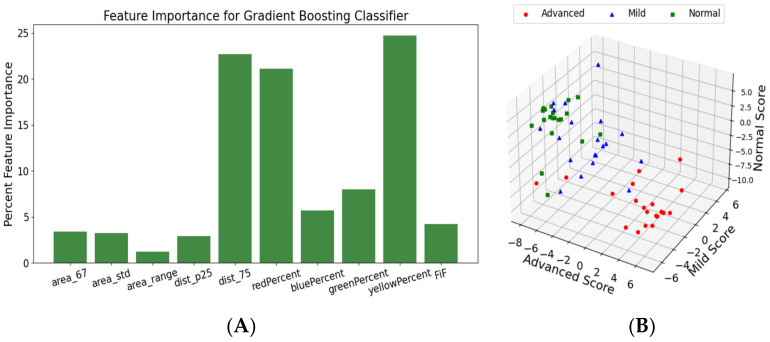
Correlation of specific features with the gradient boosting. (**A**) The importance of each parameter that contributed at least 1% to the overall score. (**B**) A 3D plot of the scores, showing how similar classes tend to cluster together.

**Table 1 diagnostics-14-01243-t001:** Description of the degrees of fibrosis based on the Ashcroft scale.

Level of Fibrosis Based on Ashcroft scale	Ashcroft Scale Score	OCT Score Based on Morphology Distortion and Birefringence Scale of Invasion
**Absent**: Normal lung	0	0: Low to moderate birefringence, thin alveoli walls
**Incipient:** Minimal fibrosis thickening of the walls of alveoli and bronchioles	1–2	1: Presence of small areas with moderate birefringence and visible thickening of the alveoli walls
**Moderate:** Moderate thickening of the walls of alveoli and bronchioles	3–5	2: Presence of large areas with increased birefringence, major thickening f the alveoli walls and reduction in the size and number of alveoli
**Severe:** Severe distortion of the lung with large masses going to total obliteration	6–8	3: Presence of massive areas with increased birefringence, severe thickening of the alveoli walls and severe reduction of the size and number of alveoli

**Table 2 diagnostics-14-01243-t002:** Balancing of the dataset.

	Number of Samples		
	Original Dataset (n = 16)	Split Dataset (n = 54)	Split Dataset + SMOTE (n = 60)
Advanced Fibrosis	7	20	20
Mild Fibrosis	6	20	20
Normal	3	14	20

**Table 3 diagnostics-14-01243-t003:** Model validation.

	Precision	Recall	Specificity	F1-Score
Advanced Fibrosis	0.94	0.85	0.98	0.89
Mild Fibrosis	0.65	0.65	0.85	0.5
Normal	0.73	0.80	0.85	0.76

## Data Availability

Data supporting reported results are considered proprietary to Physical Sciences and cannot be released without signing a confidentiality agreement. To gain access to the GitHub repository containing the code, raw images, and annotated images, please contact Dr. Nicusor Iftimia.

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
