# Peer review of "Assessing Lung Fibrosis with ML-Assisted Minimally Invasive OCT Imaging"

_diagnostics, 2024, doi:10.3390/diagnostics14121243_

Round 1

Reviewer 1 Report

Comments and Suggestions for Authors

This manuscript reports on an automated classification of lung fibrosis enabled by endoscopic PS-OCT system and ML classifier. The authors employed a custom-built PS-OCT system to image various legions in the lung of rat models, and developed a dedicated ML classifier for high-accuracy, high precision classification of lung fibrosis. Overall, the manuscript is well written, with detailed descriptions on then employed PS-OCT instrument, animal models, and ML classifiers. Some experimental results seem to convincingly validate the proposed method. However, there are several concerns that need to be addressed, before this reviewer can recommend its acceptance for publication.

- Figs. 5-7, the authors presented the birefringence OCT images, with colorbars scaled from “low” to “high”. It is strongly suggested to include quantitative values on the presented birefringence maps. For instances, many people in the OCT community use double-pass phase retardation or degree of polarization uniformity to quantitatively present birefringence information of tissues. Inclusion of this information will help the readers appreciate the changes of birefringence information in different stages of fibrosis, thereby understanding the significance and clinical utility of the PS-OCT images.

- One of the major concerns of this reviewer lies in the small number of images to train and validate the ML classifier. While it is understood that the large number of the datasets may not be readily obtained, validity and detection performance of the ML claissifer is difficult to assess, given such a small number of images. Could the authors use any kind of data augmentation schemes to increase the number of datasets?

- In Fig. 9, the authors presented filtering results for PS-OCT images using HSV codes. However, colorization of the birefringence OCT images based on the HSV level seems arbitrary, at least for the present form. Could the authors elaborate on the rationale of using such HSV levels for colorization? It is likely that colorization with different HSV levels change the ML detection performance.

- The reviewer finds it difficult to understand the significance of PS OCT images for this application. It seems that various structural information that can be obtained with conventional OCT systems, may be sufficient for fibrosis classification, if the ML classifier is trained with a large dataset, properly. Did the inclusion of PS OCT images in the ML training help improve the classification performance?

- The OCT images are highly influenced by the speckle noise, and the image quality is believed to affect the classification performance. Did the authors employ any kind of speckle reduction methods?

Author Response

Dear Reviewer,

Thank you so much for your detailed review.

Please find included our responses to your comments and suggestions.

Best regards,

                          Dr. N. Iftimia

Reviewer 1:

This manuscript reports on an automated classification of lung fibrosis enabled by endoscopic PS-OCT system and ML classifier. The authors employed a custom-built PS-OCT system to image various legions in the lung of rat models and developed a dedicated ML classifier for high-accuracy, high precision classification of lung fibrosis. Overall, the manuscript is well written, with detailed descriptions on the employed PS-OCT instrument, animal models, and ML classifiers. Some experimental results seem to convincingly validate the proposed method. However, there are several concerns that need to be addressed before this reviewer can recommend its acceptance for publication.

C1: Figs. 5-7, the authors presented the birefringence OCT images, with color bars scaled from “low” to “high”. It is strongly suggested to include quantitative values on the presented birefringence maps. For instance, many people in the OCT community use double-pass phase retardation or degree of polarization uniformity to quantitatively present birefringence information of tissues. Inclusion of this information will help the readers appreciate the changes of birefringence information in different stages of fibrosis, thereby understanding the significance and clinical utility of the PS-OCT images. 

R1: We appreciate this comment. In fact, we measured the double pass phase retardance and figures have been updated as suggested.

 C2:  One of the major concerns of this reviewer lies in the small number of images to train and validate the ML classifier. While it is understood that the large number of the datasets may not be readily obtained, validity and detection performance of the ML classifier is difficult to assess, given such a small number of images. Could the authors use any kind of data augmentation schemes to increase the number of datasets? 

R2: SMOTE sampling has been adopted to increase the number of training samples. SMOTE is an approach for data augmentation to generate data based on nearest neighbors.  

 C3: In Fig. 9, the authors presented filtering results for PS-OCT images using HSV codes. However, colorization of the birefringence OCT images based on the HSV level seems arbitrary, at least for the present form. Could the authors elaborate on the rationale of using such HSV levels for colorization? It is likely that colorization with different HSV levels changes the ML detection performance. 

R3: The color mapping in the RGB channels was directly tied to the polarization change (phase retardance). Given that this mapping of the color is consistent with the data measured for our system, the results were consistent. The use of the color, as a direct measure of the phase retardance, was a robust metric for the ML algorithm.  This metric can be applied to any PS OCT data set, if the color is a robust representation of the phase retardance.

 C4:  The reviewer finds it difficult to understand the significance of PS OCT images for this application. It seems that various structural information that can be obtained with conventional OCT systems may be sufficient for fibrosis classification, if the ML classifier is trained with a large dataset, properly. Did the inclusion of PS OCT images in the ML training help improve the classification performance? 

R4:  Indeed, the use of the polarization information helped to improve classification results. Figure 15 clearly shows that the gradient boosting algorithm for the classification was highly utilizing the polarization data in terms of yellow and red percentages. PS OCT imaging is largely used for assessing IPF:  BMJ Open Respir Res. 2023; 10(1): e001628.; https://www.atsjournals.org/doi/10.1164/rccm.202112-2832LE

 C5: The OCT images are highly influenced by the speckle noise, and the image quality is believed to affect the classification performance. Did the authors employ any kind of speckle reduction methods? 

R5:  Indeed, speckle noise can affect the ability to detect alveoli and accurately measure wall thickness. There are multiple techniques to reduce speckle artifacts. In this preliminary study we did not apply any speckle noise reduction method. However, we applied a filtering method on the calculated parameters based on the known speckle size and eliminated data where wall thickness was within the size of the speckle noise, measured as ~ 5 pixels in our images. 

Reviewer 2 Report

Comments and Suggestions for Authors

This article proposes real-time analysis of lung tissue morphology to determine the presence and level of invasiveness of idiopathic lung fibrosis (ILF) using Machine Learning techniques. There are few observations/suggestions to the authors.

1. Abstract is well drafted. Authors need to include the contributions of the proposed work at the end of the Introduction section.

2. The article does not provide any literature review. Authors must include a section (after the introduction section) named Related Work and must include the methodologies involved in solving the similar problem by various researchers. Authors also needs to bring out the literature gap and mention the same as a concluding statement in the Related Work section.

3. Authors have included the architectural diagrams for the medical instruments. Authors are suggested to include a flow/architecture diagram for classifying/identifying the ILF. The methodology may include the various stages like data collection, pre-processing (noise removal/augmentation/segmentation etc), Feature extraction and building the ML model by specifying the training and testing samples from the data set.

4. Authors are advised to increase the number of samples using augmentation technique and balance the data set.

5. A comparative analysis with existing work must be shown.

6. Why authors have not considered the CNN model? Authors are suggested to include the CNN and related models and compare the results in terms of evaluation metrics.

Overall evaluation:

This article proposes real-time analysis of lung tissue morphology to determine the presence and level of invasiveness of idiopathic lung fibrosis (ILF) using Machine Learning techniques. The article may be accepted for the possible publication only after incorporating the suggestions provided.

Author Response

Dear Reviewer,

Thank you so much for your detailed review.

Please find included our responses to your comments and suggestions.

Best regards,

                          Dr. N. Iftimia

Reviewer 2: 

C1. Abstract is well drafted. Authors need to include the contributions of the proposed work at the end of the Introduction section.

R1:  The contribution of the proposed work was included at the end of the introduction section:

In this paper we report a simpler procedure for lung fibrosis assessment based on a percutaneous approach, which enables access to any area of the lung. In contrast to the reported OCT systems, our system uses a short probe (~7 inches in length) with a diameter of 0.9 mm, which can be placed in any region of interest within the lung under US or CT guidance. The use of a small diameter probe significantly minimizes biopsy complications by comparison to the wedge biopsy approach. The minimally invasive OCT probe enables the collection of high-quality psOCT images, which can be analyzed with a custom ML-based algorithm to assess IPF presence and level of severity. We performed a preliminary evaluation of this approach in a rat model of lung fibrosis. The overall performance of the ML model was similar to that of the humans performing the same classification tasks. Specifically, tissue segmentation was excellent, closely mimicking ground truth provided by the human annotations, while >89% specificity was obtained for tissue type classification.”

C2. The article does not provide any literature review. Authors must include a section (after the introduction section) named Related Work and must include the methodologies involved in solving a similar problem by various researchers. Authors also need to bring out the literature gap and mention the same as a concluding statement in the Related Work section.

R2: A related work paragraph has been added. Typically, the related work is included in the introduction.

“Polarization sensitive OCT (PS-OCT) has been extensively used for assessing lung fi-brosis. It is used to assess excessive collagen accumulation and thus the identification and quantification of fibrosis. A recent study performed by M. Vaselli et al.22, has shown that ILD features were reliably identified with PS-OCT imaging. Microscopic ILD features were identified on both in vivo and ex vivo PS-OCT images. T. Soldati et al.23 has also shown that minimally invasive PS-OCT is a safe imaging technique to detect and quantify pul-monary fibrosis. In 49 out of 55 imaged cases, parenchymal birefringence was quantified, ranging from a mean fibrosis score of 2.54% (no to minimal fibrosis) to 21.01% (extensive fibrosis). Nandy et al.24 demonstrated that PS-OCT can reliably differentiate between inter-stitial pneumonia (UIP) and a non-UIP histopathological pattern in fibrotic lung disease. In a 27-patient study they demonstrated a 100% sensitivity and specificity for diagnosing a UIP histopathological pattern. Hariri et al. 5 have also shown that PS-OCT can provide reliable measurement of birefringent fibrosis and total collagen content.”

C3. Authors have included the architectural diagrams for the medical instruments. Authors are suggested to include a flow/architecture diagram for classifying/identifying the ILF. The methodology may include the various stages like data collection, pre-processing (noise removal/augmentation/segmentation etc), Feature extraction and building the ML model by specifying the training and testing samples from the data set.

R3:  Data segmentation section presents the steps taken to process the data. In our opinion, a diagram will be redundant.

C4. Authors are advised to increase the number of samples using augmentation technique and balance the data set. 

R4:  SMOTE was utilized to balance the dataset and expand the dataset. A follow-up paper will be prepared with larger data sets, subject to receiving additional funding.

C5. A comparative analysis with existing work must be shown.

R5:  A few references have been added and a discussion of the related work has been added as well.

C6. Why authors have not considered the CNN model? Authors are suggested to include the CNN and related models and compare the results in terms of evaluation metrics. 

R6: We compared results of 5 different classifier approaches in Figure 13. Deep learning approaches like a CNN, RCNN, or vision transformers would require a much larger dataset to train a model from the ground up. Deep learning models are blackbox, where the effects of different parameters cannot be clearly defined. The selected models are more interpretable, as shown in Fig 15. Ultimately, we wanted to be able to verify that the developed algorithm had a clear relationship to the known physics/biology of the data (birefringence, alveoli size, wall size, etc.) 

Overall evaluation:

This article proposes real-time analysis of lung tissue morphology to determine the presence and level of invasiveness of idiopathic lung fibrosis (ILF) using Machine Learning techniques. The article may be accepted for possible publication only after incorporating the suggestions provided.

R: We appreciate the positive comment.

Reviewer 3 Report

Comments and Suggestions for Authors

In the paper under review authors report an OCT-based procedure for lung fibrosis assessment using a percutaneous approach, which enables access to any area of the lung. In contrast to the conventional OCT systems, their system uses a short optical probe with a small diameter (of 0.9 mm), which can be placed in any region of interest within the lung. Also, new analysis and machine learning software was developed to interpret imaging findings and estimate the level of fibrosis invasion: mild, moderate, and severe in comparisson to the well-established Ashcroft scale. However, only 16 animals were used in the study, which is not enough for a good machine learning. In addition, the authors use 4 levels of classification of the severity of fibrosis as opposed to the Ashcroft scale of 8 levels. It reduces diagnostic efficiency. Accordingly, this should be reflected in the discussion and this reduces the total scientific significance of the presented results. These results can for now be concidered only as preliminary. Nevertheless, it is proposed to publish the article with minimal revisions, because the paper  describes an interesting approach and can be seen as the initial stage of ongoing research. In addition a minor correction is required: In Fig. 9, 10, 12, 13 text and numbers are too gray, small and hard to see. It should be improved.  

Author Response

Dear Reviewer,

Thank you so much for your review.

Please find included our responses to your comments and suggestions.

Best regards,

                          Dr. N. Iftimia

Reviewer 3:

C: In the paper under review authors report an OCT-based procedure for lung fibrosis assessment using a percutaneous approach, which enables access to any area of the lung. In contrast to the conventional OCT systems, their system uses a short optical probe with a small diameter (of 0.9 mm), which can be placed in any region of interest within the lung. Also, new analysis and machine learning software was developed to interpret imaging findings and estimate the level of fibrosis invasion: mild, moderate, and severe in comparison to the well-established Ashcroft scale. However, only 16 animals were used in the study, which is not enough for good machine learning. In addition, the authors use 4 levels of classification of the severity of fibrosis as opposed to the Ashcroft scale of 8 levels. It reduces diagnostic efficiency. Accordingly, this should be reflected in the discussion, and this reduces the total scientific significance of the presented results. These results can for now be considered only as preliminary. Nevertheless, it is proposed to publish the article with minimal revisions, because the paper describes an interesting approach and can be seen as the initial stage of ongoing research. In addition, a minor correction is required: In Fig. 9, 10, 12, 13 text and numbers are too gray, small and hard to see. It should be improved.

A: We appreciate the comments and the suggestions. The discussion section has been updated to further reflect current limitations.

“The novelty of this work should not be understated, especially that simpler and easier to fabricate OCT probes than those previously reported can be used to provide reliable PS-OCT images and assess fibrosis invasion status. Furthermore, this is the first attempt of using ML to automatically analyze PSOCT images and assess IPF severity. As far as we know, we are the first to use OCT as the driving classifier, and in contrast, are attempting to create an algorithm that utilizes in-situ imaging. Additionally, while other works have also found that an increase in birefringence corresponds with a worse fibrotic disease state22, such other studies have not built a machine learning model that could aid in the diagnosis of new IPF cases. Nevertheless, there is a long way ahead until a technology like this could be clinically translated. A finer scale of fibrosis invasion status is needed to ful-ly comply with the Ashcroft scale, while the ML approach used in this study is very pre-liminary and needs further refinement.”

Figs. 9, 10, 12, and 13 have been updated with more readable text and numbers.

Round 2

Reviewer 1 Report

Comments and Suggestions for Authors

The authors well addressed the concerns raised by this reviewer. As such, this reviewer supports the publication of this manuscript.

Author Response

Thank you!

Reviewer 2 Report

Comments and Suggestions for Authors

Authors have incorporated all the suggestions provided. The article may be accepted for the possible publication.

Author Response

Thank you!